# Electroassisted Incorporation of Ferrocene Within Sol–Gel Silica Films to Enhance Electron Transfer—Part II: Boosting Protein Sensing with Polyelectrolyte-Modified Silica

**DOI:** 10.3390/molecules30153246

**Published:** 2025-08-02

**Authors:** Rayane-Ichrak Loughlani, Alonso Gamero-Quijano, Francisco Montilla

**Affiliations:** 1Departamento de Química Física, Instituto Universitario de Materiales de Alicante (IUMA), Universidad de Alicante, Carretera San Vicente s/n, 03690 Alicante, Spain; rayane.loughlani@univ-biskra.dz (R.-I.L.); alonso.gamero@icp.csic.es (A.G.-Q.); 2Department of Sciences Matter, Faculty of Exact Sciences, University of Biskra, P.O. Box 145, Biskra 07000, Algeria; 3Instituto de Catálisis y Petroleoquímica—Consejo Superior de Investigaciones Científicas (ICP—CSIC), Calle de Marie Curie 2, 28049 Madrid, Spain

**Keywords:** hybrid silica functionalization, ferrocene, electroassisted accumulation mediated electron transfer, cytochrome c

## Abstract

Silica-modified electrodes possess physicochemical properties that make them valuable in electrochemical sensing and energy-related applications. Although intrinsically insulating, silica thin films can selectively interact with redox species, producing sieving effects that enhance electrochemical responses. We synthesized Class I hybrid silica matrices incorporating either negatively charged poly(4-styrene sulfonic acid) or positively charged poly(diallyl dimethylammonium chloride). These hybrid films were deposited onto ITO electrodes and evaluated via cyclic voltammetry in aqueous ferrocenium solutions. The polyelectrolyte charge played a key role in the electroassisted incorporation of ferrocene: silica-PSS films promoted accumulation, while silica-PDADMAC films hindered it due to electrostatic repulsion. In situ UV-vis spectroscopy confirmed that only a fraction of the embedded ferrocene was electroactive. Nevertheless, this fraction enabled effective mediated detection of cytochrome c in solution. These findings highlight the crucial role of ionic interactions and hybrid composition in electron transfer to redox proteins, providing valuable insights for the development of advanced bioelectronic sensors.

## 1. Introduction

Silica materials exhibit unique properties due to the size and acid–base characteristics of their porous structure [1,2]. Whilst the pore charge can be precisely tuned by adjusting the bulk pH relative to the isoelectric point of ca. 2, the pore size can be tailored through sol–gel methods and the critical micelle concentration of surfactants [3,4]. Crucially, the versatility of silica materials is further enhanced by incorporating organic modifiers. For instance, Class I hybrids are synthesized by incorporating polymeric surfactants into the pores, where non-covalent bonds form between the silica matrix and the modifier. Conversely, Class II hybrids, typically based on organo-modified silanes (ORMOSILs), achieve a robust covalent linkage between the organic component and the polysiloxane network [5,6,7]. Despite their insulating properties, sol–gel silica materials have been widely used in electrochemistry [8,9,10,11]. High adsorption capacity, biocompatibility, and easy processing with exceptional chemical properties make silica-based materials highly suitable for electrode modification with precise control over electrochemical performance [12,13,14,15].

The most straightforward application of silica-modified electrodes in analytical electrochemistry is the electroassisted accumulation (pre-concentration) of electroactive analytes within the films [16,17]. The accumulation of the analyte can occur spontaneously (physisorption) or be electroassisted by an external step potential (electrosorption). Whether spontaneous or electroassisted, this pre-concentration step enhances the sensitivity and response time in detecting analytes, and thus it has been broadly employed in electroanalytical chemistry.

Different approaches may be used to carry out the electrosorption within the films. For instance, a double potential step approach can be employed, where a preconcentration step potential is applied for a short period, followed by a second potential step during which the electrochemical detection occurs. Furthermore, electrosorption can also occur under potentiodynamic conditions using cyclic voltammetry, where the analyte accumulates during each voltammetric sweep. However, the evolution of the voltammetric signals will likely depend on the affinity between the analyte and the net charge of the silica pores, as well as the pH [17]. The limitation regarding pH could be addressed by incorporating charge-selective polyelectrolytes with high or low pKa into the silica network, resulting in a positively or negatively charged Class I hybrid film, regardless of the bulk pH of the solution [18,19].

Herein, to modify the net charge of the silica pores, we chose both anionic and cationic polyelectrolytes, which will be integrated into the silica films to create Class I silica films. Poly (diallyl dimethylammonium chloride) (PDADMAC, pK_a_ = 10.5 [20]), is a cationic polyelectrolyte, and poly(4-styrene sulfonic acid) (PSS, pK_a_ = 1.5 [21]), is an anionic polyelectrolyte [22]. These Class I films were tested in an aqueous ferrocenium solution, and the electroassisted accumulation of ferrocene was evaluated.

Previous research has shown that increasing the hydrophobicity of silica pores with methyl groups, or growing conducting polymers like PEDOT, improves electron transfer between the electrode and proteins [20,21]. Therefore, as proof of concept, the performance of ferrocene-doped Class I films was tested in the presence of a redox-active protein such as Cytochrome c (Cyt c). Cyt c is an ideal biological redox probe since, *(i)* it presents a globular shape with a 3 nm diameter [23], (*ii*) it has a positive net charge of +9 at neutral pH [24], and (*iii*) it presents a reversible electrochemical response.

Building on our findings, we have explored how the ionic nature of the silica-modified film and the incorporation of ferrocene species could determine the redox response of cytochrome c in solution. We have synthesized silica-PSS and silica-PDADMAC thin films that were further functionalized with ferrocene species through electroassisted accumulation. These films were characterized by cyclic voltammetry and in situ UV-vis spectroscopy.

While much of the previous research has focused on the development of Class II hybrid silica materials, typically involving covalent modification through organosilane chemistry or advanced click-chemistry approaches [25,26,27,28], the present study offers a simpler and more accessible alternative. Conventional strategies for ferrocene incorporation generally require multistep syntheses and covalent grafting techniques to ensure stable immobilization and minimize leaching into the electrolyte solution [25,26,27,28]. In contrast, this work employs a single-step electroassisted method to incorporate ferrocene into Class I hybrid silica films formed via non-covalent interactions with charged polyelectrolytes. Moreover, this study aims to advance the field by systematically investigating the influence of polyelectrolyte identity (PSS vs. PDADMAC) on the electroassisted incorporation and retention of ferrocene species, as well as their capacity to mediate electron transfer to redox proteins such as cytochrome c.

## 2. Results and Discussion

### 2.1. Electrochemical Characterization of Hybrid Silica-Modified Electrodes

The electrochemical performance of ITO/silica-PSS and ITO/silica-PDADMAC was evaluated by cyclic voltammetry in a 0.3 mM FcPF_6_ Trizma aqueous solution. A schematic illustration of the electroassisted accumulation process of ferrocene within the sol–gel hybrid silica film is provided in (Appendix A), offering a visual representation of how ferrocene species are retained within the hybrid silica film structure.

Figure 1a shows the ferrocenium redox response using a silica-PSS-modified electrode.

The first scan (dashed line) displays two overlapping anodic peaks centered at 0.94 V and 1.35 V. These peaks are attributed to mass transport limitations of ferrocenium cations within the silica-modified films [26]. The first oxidation peak is attributed to the oxidation of ferrocene species retained in the inner layers of the silica matrix. The second oxidation peak is assigned to the oxidation of ferrocene species retained in the outer layers of the silica matrix [17]. The presence of two oxidation peaks is related to the compensation of positive charges within the silica film induced by ferrocenium cations [25,26]. During the backward scan, the counter process (reduction of ferrocenium cations to neutral ferrocene) was observed as a single cathodic peak centered at 0.73 V. The cathodic and anodic peak intensities increased stepwise with the repetitive cycling (see the 10th and 60th cycles in Figure 1a), indicating a continuous accumulation of ferrocenium species within the silica-PSS film. After 20 cycles, the two anodic peaks merged, meaning that both processes overlapped in a single bell-shaped peak, shown by dotted line in Figure 1a. The voltammogram reached a steady state after 60 cycles, as shown by the solid line in Figure 1a.

The peak-to-peak separation of the stabilized voltammogram was 0.61 V, and the half-wave potential was 1.04 V. Here, the large peak-to-peak separation could be attributed to a sluggish heterogeneous electron transfer between the electrode and the redox probe. The anodic and cathodic peak intensities obtained after 60 scans were around 207 µA and −282 µA, respectively. This corresponds to an anodic/cathodic peak ratio of 0.73.

It is worth noting that this anodic/cathodic peak ratio is higher than that obtained with silica polyelectrolyte-free modified electrodes [17], which indicates an enhanced accumulation of ferrocenium cations near the electrode surface. This enhancement was due to the electrostatic attraction between the ferrocenium species and the sulfonic groups present in the PSS polymer chain. The presence of negatively charged sulfonate groups (SO_3_^−^) in the PSS film likely facilitates positive charge compensation during the oxidation of ferrocene. When ferrocene is oxidized to ferrocenium, the positive charge excess is locally balanced by the available sulfonate groups within the film, thereby decreasing the charge transfer limitation within the film and facilitating the oxidation process.

The pore size of silica-PSS was obtained by applying the DFT method to the adsorption isotherm (See Appendix A). The analysis of pore size distribution in the silica-PSS sample indicates the presence of both micropores, with a diameter of 1.46 nm, and mesopores, with a diameter of 6.5 nm. Notably, these values are smaller by 0.04 nm and 0.5 nm, respectively, compared to those reported for polyelectrolyte-free silica. [17] This slight reduction in both pore types indicates that the incorporation of PSS alters the internal structure of the silica matrix. It is likely that PSS interacts with the silica network during the sol–gel process, leading to decreased pore dimensions and modified porosity.

Figure 1b shows the ferrocenium redox response using a silica-PDADMAC-modified electrode. The first voltammetric scan also showed two anodic peaks, centered at 0.96 V and 1.19 V, respectively. A single cathodic peak at 0.73 V was observed during the backward scan. The presence of the two oxidation peaks was attributed to mass transport limitations, as discussed *vide supra* for silica-PSS-modified electrodes. A single bell-shaped peak was observed after the 85th repetitive voltammetric scan.

The cathodic and anodic peak intensities exhibited a stepwise increase with repetitive cycling, as observed in the CV cycles from 10th to 85th in Figure 1b. The stabilized CV cycle showed a peak-to-peak separation of 0.63 V, and the half-wave potential was 1.03 V. The anodic and cathodic peak intensities after 85 scans were approximately 125 µA and −219 µA, respectively. The anodic/cathodic peak ratio was 0.57. The latter indicates unbalanced concentrations of ferrocene and ferrocenium at the interface of the silica-PDADMAC-modified electrode. Specifically, the concentration of ferrocenium cations was higher than that of neutral ferrocene, which was expected given that the electrode was immersed in a solution containing only ferrocenium cations. It is worth noting that the repulsion of ferrocenium ions from silica film containing the positively charged polyelectrolyte will produce a lower initial concentration of ferrocenium ions near the electrode surface, leading to a decrease in the concentration of ferrocene. This imbalance in the concentration of redox species can affect electron transfer kinetics, as reported in a previous work [17].

The pore characteristics of the silica-PDADMAC film were also evaluated using the DFT model applied to the adsorption isotherm (see Appendix A). The resulting distribution showed the presence of micropores with a diameter of 1.56 nm and mesopores measuring 7.7 nm. Compared to pure silica [17], the micropores were slightly wider by 0.06 nm, and the mesopores by 0.7 nm, indicating that PDADMAC incorporation leads to a more open porous structure. This structural expansion may be attributed to electrostatic interactions between the cationic PDADMAC and the negatively charged silica walls during the sol–gel process. Furthermore, we carried out experiments at varying scan rates, confirming the presence of two different kinetics of heterogeneous electron transfer attributed to mass transport limitations within the hybrid silica films (see Appendix A). For both silica-PSS and silica-PDADMAC-modified electrodes, the cyclic voltammograms recorded at low scan rates (10–20 mV s^−1^) displayed a single anodic peak with a bell-shaped profile. This feature is indicative of ferrocene species either adsorbed at the electrode interface or slowly diffusing through the porous silica matrix. Under these conditions, the diffusion layer is sufficiently thick to mask mass transport limitations within the films. As the scan rate increased to 50 mV s^−1^, a noticeable shoulder developed on the anodic wave for both hybrid silica-modified electrodes, signaling the emergence of a secondary oxidation process, most likely involving ferrocene species confined deeper in the silica films. At higher scan rates (500 and 1000 mV s^−1^), pronounced peak broadening and distortion were evident in both hybrid silica-modified electrodes, reflecting growing mass transport limitation and slower heterogeneous electron transfer kinetics across the hybrid silica films.

The capacity to accumulate Ferrocene within silica-PSS-modified electrodes was higher than using silica-PDADMAC-films (almost two-fold). Furthermore, we observed that the addition of a cationic polyelectrolyte still permitted the accumulation of ferrocene species within the silica film, meaning that a partial neutralization of the negatively charged pores of the silica walls occurred by the presence of the positively charged PDADMAC polyelectrolyte.

It is worth mentioning that in the case of PDADMAC-modified electrodes, the first anodic peak showed a low current, while the subsequent oxidation peak showed a higher intensity. This contrasted with the voltammograms recorded with silica-PSS films, where the first oxidation peak displays the highest current intensity. This divergence can be attributed to the repulsion between positive charge species and the restricted charge compensation within the inner layer of the films during ferrocene oxidation.

In silica-PSS films, the negatively charged sulfonate groups interact with the ferrocenium ions, influencing their distribution and promoting their accumulation primarily in the inner layers of the film. This leads to a dominant first oxidation peak because charge compensation occurs efficiently from the start of the oxidation reaction. Over several voltammetric cycles, ferrocenium species keep accumulating, gradually fading the difference between the inner and outer layer oxidation peaks.

In silica-PDADMAC films, the situation is different due to electrostatic repulsion between ferrocenium ions and the positively charged PDADMAC. This repulsion hinders ferrocenium accumulation in the inner layers, resulting in a weaker first oxidation peak. However, as oxidation progresses and charge redistribution occurs, ferrocenium species accumulate more efficiently in the outer layers of the film, where there is less repulsion. This results in a more pronounced second oxidation peak, which remains dominant throughout the cycling process.

In summary, silica-PSS presents a larger accumulation capacity for ferrocene compared to the PDADMAC-modified electrode, as quantified by the current reduction peaks in the voltammogram (−282 µA for PSS vs. −219 µA for PDADMAC).

To evaluate how stable ferrocenium is retained within the hybrid silica films, electrodes were removed from the electrochemical cell and thoroughly rinsed with abundant ultrapure water (See Appendix A). The electrodes were characterized by cyclic voltammetry in a ferrocenium-free solution. A schematic representation of the electroassisted desorption process is provided in Appendix A, illustrating the gradual release of ferrocene species from the hybrid films during potential cycling.

Figure 2 shows repetitive cyclic voltammograms obtained with (a) PSS and (b) PDADMAC Class I films after the ferrocene accumulation stage. The first notable aspect of these voltammograms, in comparison to the final voltammograms of Figure 1, was the lowering of the intensity of the redox peaks (as expected by the rinsing stage) and the modification of the shape of the voltammogram.

During the first voltammetric scan, both electrodes presented a single ferrocene-ferrocenium diffusional-like oxidation peak, in contrast to the bell-shaped peak previously observed in Figure 1 when ferrocenium was dissolved in the aqueous buffer solution. We consider this peak to correspond to the confined ferrocene species’ response within the silica films’ inner layers, while loosely retained ferrocene was removed during the rinsing.

Considering the first PSS-functionalized silica (Figure 2a), we observed a continued decrease in the anodic and cathodic peak intensities during the voltammetric cycling due to the partial leaching of the remaining ferrocene. After 30 cycles, a stable signal was reached with a peak-to-peak separation equal to 215 mV and a reduction peak current intensity of −14 µA (solid line in Figure 2a), that is, only the 5% of the current detected after the accumulation step (final scan of Figure 1a). We determined the apparent ferrocene concentration within the silica film making use of the Randles–Sevcik equation [29,30]:(1)$${I_{p}\;=\;\left(269\times 10^{5}\right)\;n}^{3/2}AC\;D^{1/2}\nu^{1/2}$$
where *I_p_* (*A*) is the peak current, υ is the scan rate (V s^−1^), *n* is the number of electrons transferred, *A* the electrode surface area (cm^2^), D the diffusion coefficient of the analyte (cm^2^ s^−1^), and C the concentration of the analyte (mol cm^−3^) in the bulk solution.

We assume a value for the diffusion coefficient of 1.2 × 10^−6^ cm^2^ s^−1^ as determined in the trizma buffer solution [17]. It should be noted that this value represents an approximation and not the true diffusion coefficient within the film. Therefore, the calculated concentrations are not absolute but serve as relative estimates to compare the retention of ferrocene in the different hybrid silica films. In this context, the peak current value (*I_p_*) was obtained from the reduction peak of the stabilized cyclic voltammogram. By applying the Randles–Sevcik equation with the known parameters (scan rate, electrode area, and estimated diffusion coefficient), we calculated the apparent concentration of electroactive ferrocene. Based on this analysis, the apparent concentration of electroactive ferrocene in the silica-PSS film was approximately 0.03 mM.

Similar experiments performed with silica functionalized with PDADMAC are shown in Figure 2b. In the first scan towards positive potentials, a single peak at 0.96 V corresponds to the oxidation of the confined ferrocene species within the inner layers of the silica-PDADMAC film. During the backwards scan, the counter process was observed as a cathodic peak centered at 0.74 V. The anodic and cathodic peak intensities decreased stepwise throughout the voltametric cycling due to the leaching of the ferrocene species into the solution. After 50 cycles a stable signal was reached. The stabilized cyclic voltammogram shows a reduction peak with a current intensity of −46 µA. The apparent ferrocene concentration determined by Randles–Sevcik analysis was estimated to be 0.08 mM; that is more than double the concentration determined for silica-PSS films.

The lower retention of ferrocene species in silica-PSS compared to silica-PDADMAC is notable and can be explained by differences in electrostatic interactions and film stability.

In silica-PSS films, the negatively charged sulfonate groups should facilitate the initial accumulation of ferrocenium ions; however, weaker retention forces and the high hydrophilicity of the film contribute to increased leaching during rinsing and cycling. Conversely, in silica-PDADMAC films, the positively charged matrix initially hinders ferrocenium incorporation due to electrostatic repulsion, but once retained, stronger affinity due to electrostatic and hydrophobic interactions minimize ferrocenium loss. As a result, the final concentration of ferrocene species within silica-PDADMAC films is higher than that observed in silica-PSS films.

The results indicate that the relevant interaction between the encapsulated ferrocenium/ferrocene couple and the hybrid silica is not the electrostatic one.

These findings can be further rationalized by distinguishing the dominant interaction at each stage of ferrocene processing within the films. During the initial accumulation step, electrostatic forces govern the incorporation of ferrocenium ions: the negatively charged sulfonate groups in silica-PSS films attract ferrocenium ions efficiently, while the electrostatic repulsion in PDADMAC-modified films limits their accumulation. This explains the initially lower voltammetric response in PDADMAC-modified films (Figure 1b). However, once incorporated, neutral ferrocene exhibits higher affinity for the hydrophobic PDADMAC matrix, likely due to favorable hydrophobic interactions with the polymer backbone. This affinity improves ferrocene retention during rinsing and voltammetric cycling, as evidenced by the higher stabilized current intensities and calculated electroactive concentrations in Figure 2b. In contrast, silica-PSS films, though efficient in accumulating ferrocenium, exhibit lower retention due to their hydrophilic environment, which facilitates ferrocene leaching. Therefore, while electrostatics dominate the initial loading, retention is predominantly governed by hydrophobic interactions within the film matrix.

The backbone of PDADMAC is relatively hydrophobic, and ferrocene in its neutral form is also moderately hydrophobic. Such interactions may favor the adsorption or retention of ferrocene within the hydrogel layer, potentially influencing its diffusion dynamics and redox behavior, especially under conditions where electrostatic effects are minimized or screened by counterions.

It is worth noting that the ITO electrode modification with silica-PSS and silica-PDADMAC significantly enhanced the reduction peak intensity by approximately 5-fold and 3-fold, respectively, compared to bare ITO [17]. Additionally, the hybrid silica-modified electrodes effectively prevented electrode fouling [17]. Importantly, no noticeable increase in resistance or loss of capacitive behavior was observed after surface modification, indicating that the silica layer does not hinder charge transport at the electrode interface (See Appendix A).

Figure 3 shows the in situ UV-vis spectra of Fc@hybrid silica modified-electrodes.

After ferrocenium stabilization (as described for experiments shown in Figure 2), the stabilized Fc@silica-modified electrodes were extracted from the electrochemical cell at 1.42 V to ensure that ferrocenium stays in the oxidized state within the silica. Then, the electrodes were transferred to the UV-vis spectroelectrochemical cell containing a blank solution. Figure 3a shows the response of a Fc@silica-PSS electrode for an applied potential of 1.42 V. The spectrum exhibits an absorption band centered at 618 nm attributed to the ligand-to-metal charge transfer band of ferrocenium incorporated within the silica pores [17,31]. From the absorbance, the apparent concentration of ferrocenium within the film was determined to be 1.40 mM (Figure 3a). Details on the determination of this concentration were given elsewhere [17].

We studied the electrochemical reduction process of encapsulated ferrocenium by applying a potential of +0.22 V vs. RHE. We kept this potential constant to ensure the complete reduction of electroactive ferrocenium within the film. The final stabilized spectrum obtained (dashed line in Figure 3a) shows a decrease in the intensity of the ferrocenium absorption band at 618 nm, confirming that a fraction of ferrocenium species was reduced in the film. The intensity of the absorption band allows us to determine the portion of encapsulated ferrocenium that was effectively reduced to ferrocene. The final concentration of encapsulated Fc^+^ species was still 1.17 mM. The difference between both concentrations is related to the concentration of ferrocenium that is electroactive within the film. In this case, the amount of electroactive ferrocenium was 0.23 mM, a value much higher than that determined from the voltammetric Randles–Sevcik analysis (0.03 mM).

Figure 3b shows the in situ UV-vis spectra of a Fc@silica-PDADMAC electrode. The spectrum obtained is very similar to the one obtained with Fc@silica-PSS. From the absorbance, the apparent concentration of encapsulated ferrocenium was 1.87 mM. After applying a potential of +0.22 V vs. RHE (dashed line spectrum in Figure 3b) a decrease in the ferrocenium intensity band was observed reaching a concentration of 1.76 mM. The difference between oxidized and reduced ferrocenium species corresponds to the electroactive species within the film, that was 0.11 mM closer to the voltammetric estimation (0.08 mM).

Table 1 summarizes the electroactive ferrocene species incorporated in silica films measured by both techniques. For comparative purposes, we also included the data obtained with conventional silica (with no polyelectrolyte incorporated).

Cyclic voltammetry measured a lower concentration of electroactive ferrocenium than the concentration determined by in situ UV-vis. This discrepancy is particularly apparent in the silica modified with negatively charged sulfonated moieties (PSS).

Concentration derived from CV in silica functionalized with positively charged groups (PDADMAC) closely aligns with the spectroelectrochemical results. Since the ferrocenium concentration was determined by using the Randles–Sevcik equation (Equation (1)), we considered that the diffusion coefficient of ferrocenium was similar to that in solution. However, in previous works, we demonstrated that the silica pores can effectively modulate the diffusion of charged redox probes [32]. Indeed, negatively charged redox species (like ferrocyanide species or aromatic sulfonates) show enhanced diffusivities across the SiO_2_ pores functionalized with PSS, in contrast to positively charged redox species, which show lower diffusivities [33].

Since silica is negatively charged under working conditions, adding PSS increases the number of negative charges on the silica walls. Consequently, the diffusion coefficient used to determine ferrocenium concentration is likely higher than the basal value within the silica films, resulting in an underestimation of the ferrocenium concentration in the films via voltammetry means. Table 1 also shows the concentration of ferrocene species within polyelectrolyte-free silica films [17]. Similarly, the estimation from CV measurements indicated a local concentration of 0.12 mM, which is lower than the UV-vis estimate of 0.19 mM. The difference is less pronounced when compared to Fc@silica-PSS, but it follows a similar trend as discussed for Class I films.

A different effect is inferred for Fc@silica-PDADMAC films. At neutral pH, the cationic polyelectrolyte (PDADMAC) partially neutralizes the negative charges on the silica walls, resulting in a neutral-like pore that has a less pronounced effect on the diffusion of redox species. Thus, the ferrocenium concentration measured by CV (0.08 mM) was much closer to the value obtained by UV-Vis (0.11 mM). This suggests that the discrepancy between electrochemical and spectroscopic methods could be attributed to the negative charge density within the silica pores.

The comparison between the concentration from in situ UV-vis experiments and cyclic voltammetry suggests that the diffusion coefficient of ferrocenium/ferrocene couple was underestimated in the voltammetric measurement. In previous publications, we determined that silica-PSS-modified electrodes modify the apparent diffusion coefficient of positively charged redox probes (such as Fe^3+/2+^ or dopamine). Under these circumstances, the electrostatic interactions between the positively charged species and the films result in lower diffusion coefficients, leading to retention of the probes.

However, the observed decrease in the apparent diffusion coefficient observed in the presence of PDADMAC suggests that hydrophobic interactions between ferrocene and the hybrid film are the most significant. Likely, the positively charged segments of PDADMAC interact electrostatically with the negatively charged silanol and siloxane groups (Si–OH, Si–O^−^) on the silica surface, resulting in the orientation of the hydrophobic alkyl chains of PDADMAC toward the interior of the pores. This configuration creates a hydrophobic microenvironment that favors hydrophobic interactions with ferrocene, thereby retarding its diffusion through the film [34]. These findings highlight the dominant role of hydrophobic interactions in governing the diffusion behavior of ferrocene within the hybrid films studied [35,36,37].

To investigate how confined ferrocene species improve electron transfer, we studied a redox species with restricted direct electron interaction with the electrode, such as Cytochrome c (Cyt c). Therefore, the following section discusses the interaction between confined ferrocene species within Class I silica films and Cyt c in solution. It is worth noting that the results shown in Figure 4 were obtained after determining the concentration of ferrocenium/ferrocene within the silica films after immersion in PBS, i.e., the electrochemical response of Cyt c in solution depended on the local concentration of ferrocene within the films.

### 2.2. Electrochemical Performance of Fc@ Hybrid Silica-Modified Electrodes for Electron Transfer to Cyt c

Figure 4 shows the stabilized cyclic voltammogram of Fc@hybrid silica-modified electrodes in PBS buffer solution in the presence of Cyt c. A schematic illustration of this experiment is provided in Appendix A.

For PSS-functionalized silica (Figure 4a), the voltammogram obtained in the blank solution (dashed line) shows a quasireversible signal with an oxidation peak at 1.09 V and a poorly defined reduction near 0.44 V. These peaks are attributed to the redox response of the confined ferrocene/ferrocenium species within the film. The electrochemical response of the Fc@silica-PSS electrode in a solution containing Cyt c shows well-defined redox processes attributed to the redox protein. During the forward scan, a couple of anodic peaks were recorded at 1.13 V and 0.87 V. The peak at 0.87 V corresponds to cyt c-Fe (II) oxidation, whereas the second anodic peak centered at 1.13 V corresponds to the oxidation of ferrocene species confined within the silica matrix. During the backwards scan, a single cathodic peak appeared centered at 0.44 V, indicating that ferrocenium and cyt c are electrochemically reduced at this potential. The charge under the cathodic peak increased by approximately 31 µC compared to the cathodic peak recorded in a blank solution, indicating that a mediated electron transfer occurs between the encapsulated ferrocene and cyt c species.

It should be emphasized that the electrochemical response of Cyt c on silica-PSS-modified electrodes presents a featureless redox double-layer process (See Appendix A), suggesting that there is no adsorption of Cyt c on the silica-PSS-modified electrode surface.

For silica modified with the cationic polyelectrolyte, the voltammogram of the blank solution (Figure 4b) shows a quasireversible signal of the confined ferrocene/ferrocenium species within the silica-PDADMAC film. Similar to the Fc@silica-PSS electrode, the electrochemical response of the Fc@silica-PDADMAC electrode in a Cyt c-containing solution shows distinct redox processes attributed to Cyt c. During the forward scan, an anodic peak appears at 0.91 V, with a shoulder at 0.73 V corresponding to cyt c-Fe(II). The backward scan shows a single cathodic peak at 0.44 V, indicating that both ferrocenium and cyt c are electrochemically reduced at this potential. The charge under the cathodic peak in the presence of Cyt c increased by approximately 23 µC compared to the blank solution, further confirming the electron transfer between the encapsulated ferrocene and Cyt c species.

Table 1, last column, compiles the data for the charge transfer associated with Cyt c with the different Class I silica-modified electrodes. Since Cyt c’s isoelectric point is in the range (10–10.6) [38], the presence of the PSS enhances the charge transfer compared with conventional silica film (28 µC) [17].

The observed differences in mediated electron transfer between the Fc@silica-PSS and Fc@silica-PDADMAC electrodes can be attributed to variations in surface chemistry and ferrocene retention within the hybrid films. The incorporation of PSS into the silica matrix introduces a high density of negatively charged sulfonate groups, increasing the film’s hydrophilicity and promoting electrostatic interactions with Cyt c in solution. This likely increases the local concentration of Cyt c near the film surface, thereby enhancing the frequency of productive collisions with the confined ferrocene species and facilitating more efficient mediated electron transfer. In contrast, PDADMAC-functionalized films present a more hydrophobic environment with a lower surface charge density. This modification likely reduces the local concentration of Cyt c near the electrode surface due to weaker electrostatic interactions, limiting its accessibility to the confined ferrocene species within the silica film. Additionally, potential differences in pore morphology or tortuosity introduced by PDADMAC may further hinder Cyt c diffusion or reduce its spatial proximity to electroactive sites.

Additionally, the improved response of Cyt c with the Fc@silica-PSS film compared to the Fc@silica film correlates with a higher electroactive concentration of ferrocene species confined within the Fc@silica-PSS film. The electroactive ferrocene species are critical in mediating electron transfer during the Cyt c detection process. This trend is further supported by the results for the Fc@silica-PDADMAC film, which showed a lower electroactive ferrocene concentration, correlating with a diminished Cyt c detection signal.

## 3. Materials and Methods

The following reagents were used: tetraethyl orthosilicate (TEOS, Sigma-Aldrich, Madrid, Spain, reagent grade), ferrocenium hexafluorophosphate (97%, Sigma-Aldrich), poly(diallyl dimethylammonium chloride) 20% wt. (PDADMAC, Sigma-Aldrich), and poly(4-styrene sulfonic acid) 18% wt. (PSS, Sigma-Aldrich). Cytochrome c from bovine heart (95%, Sigma-Aldrich), ethanol (Sigma Aldrich), hydrochloric acid (Merck Life Science SLU, Madrid, Spain, 37%), Trizma base (99.9%, Sigma-Aldrich), nitric acid (65%, Panreac, Madrid, Spain), potassium nitrate (Merck, 99%), potassium dihydrogen phosphate, and dipotassium hydrogen phosphate (99%) were from VWR Chemicals (Avantor, Radnor, U.S.A.). All solutions were prepared using ultrapure water (18.2 MΩ cm) obtained from an ELGA lab water Purelab system (ELGA LabWater, High Wycombe, U.K.) Phosphate buffer solution (PBS, pH 7.3) was prepared from K_2_HPO_4_ (0.15 M) and KH_2_PO_4_ (0.1 M).

Electrochemical experiments were carried out in a 3-electrode setup. The working electrodes were indium tin oxide (ITO)-coated glass substrates (SOLEMS, ITOSOL30, sheet resistance 25–35 Ω). Prior to use, the ITO glass was degreased by sonication in acetone and electro-oxidized galvanostatically at 0.1 mA cm^−2^ for 1 min in Trizma solution. A reversible hydrogen electrode (RHE) was used as the reference electrode, and a platinum wire was used as the counter electrode. The electrochemical cells were purged with nitrogen gas for 10 min before the experiments, and the nitrogen atmosphere was maintained during all the experiments. Electrochemical measurements were conducted using an ES161 eDAQ potentiostat (eDAQ PTY Ltd, Sydney, Australia) with an ED401 for data acquisition, controlled by Echart software (version 5.2).

UV-vis spectra were collected using a JASCO V-730 spectrophotometer (Jasco Inc, Easton, U.S.A) with 1 cm optical path quartz cuvettes. In situ UV-vis spectroscopy was performed using a custom-made spectroelectrochemical quartz cuvette under a nitrogen atmosphere. All measurements were carried out at room temperature.

To assess the porosity of the hybrid silica materials, nitrogen adsorption analyses were carried out at −196 °C using an automated system (Autosorb-6, Quantachrome, Boynton Beach, FL, USA). The hybrid silica was synthesized as monoliths using the sol–gel method and subsequently dried under vacuum at 100 °C for 24 h. Pore size distribution was calculated from the adsorption isotherms using the Density Functional Theory (DFT) model implemented in the Autosorb-6 software (Quantachrome).

Two stock solutions were prepared for hybrid silica film preparation:

Solution 1: The silica-polyelectrolyte precursor sol was made by mixing 3 mL of TEOS and 4.1 mL of ethanol. We added 2.9 mL of 0.01 M HCl aqueous solution containing polyelectrolyte, either 0.05 M of poly 4-styrene sulfonic acid (PSS, 9.20 g/L) or 0.05 M poly(diallyl dimethylammonium chloride, PDADMAC, 8.08 g/L). This mixture was magnetically stirred for one hour at room temperature in a sealed glass vial.

Solution 2: Trizma buffer solution (pH = 8.44) was prepared by mixing 0.1 M Trizma base with 0.1 M KNO_3_, adding concentrated HNO_3_ dropwise until the desired pH was reached.

To prepare the polyelectrolyte–silica composites, solutions 1 and 2 were mixed in a 1:1 volume ratio, respectively, in an Eppendorf vial. A volume of 40 μL of the mixture was spread onto 1.8 cm^2^ of a clean ITO electrode. After 2 to 3 min, a homogeneous and transparent silica hybrid sol–gel film was formed on the surface of the ITO electrode. Under these conditions, the resulting hybrid silica hydrogel films were estimated to be approximately 200 µm thick, ensuring uniform coverage of the ITO surface.

## 4. Conclusions

This study focused on characterizing and understanding the accumulation process of ferrocenium species within Class I silica films, along with their application in mediated electron transfer reactions to redox protein, Cyt c. The research successfully demonstrated the functionalization of the silica matrix with poly(4-styrene sulfonic acid) and poly(diallyl dimethylammonium chloride) using the sol–gel method. The silica-PSS film showed significantly higher capacity for ferrocenium species accumulation than conventional silica.

Furthermore, the findings indicate that, in addition to electrostatic interactions, hydrophobic effects significantly influence the retention of ferrocene species within the hybrid silica matrices. In particular, the hydrophobic character of the poly(diallyl dimethylammonium chloride) backbone is likely to enhance non-covalent interactions with neutral ferrocene molecules, thereby promoting their stabilization within the silica network. This effect likely contributes to the higher stability and lower leaching observed in PDADMAC-modified films compared to PSS-modified films, despite the initial electrostatic repulsion.

In situ UV-vis spectroscopy showed that only a fraction of the ferrocene species within the Fc@silica-PSS and Fc@silica-PDADMAC films displayed electroactivity. However, this fraction was still sufficient to enhance the electrochemical signal of Cyt c in solution.

The incorporation of poly(4-styrene sulfonic acid) and poly(diallyl dimethylammonium chloride) within the silica matrix modulates the affinity of the silica for Cyt c. Notably, the electrochemical signal of Cyt c was found to be enhanced in the following order: Fc@silica-PSS > Fc@silica > Fc@silica-PDADMAC.

Our findings show the potential of hybrid silica films for accumulating ferrocenium species, providing valuable insights into designing and optimizing ferrocene-hybrid silica-modified electrode materials for mediated electron transfer reactions. This could pave the way for exploring their potential applications as electron transfer mediators in bioelectrochemistry.

## Figures and Tables

**Figure 1 molecules-30-03246-f001:**
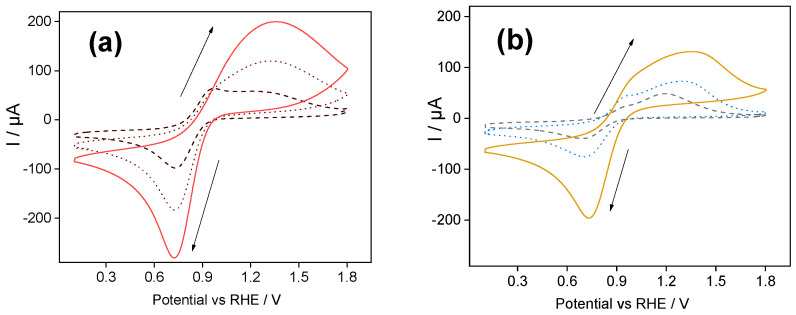
(**a**) Cyclic voltammograms of an ITO/silica-PSS-modified electrode in 0.3 mM FcPF_6_ solution (Trizma buffer) at 100 mV s^−1^: Dashed line—1st scan; dotted line—10th scan; solid line—60th scan. (**b**) Repetitive cyclic voltammograms for an ITO/silica-PDADMAC-modified electrode in 0.3 mM FcPF_6_ solution (Trizma buffer) at 100 mV s^−1^: Dashed line—first scan; dotted line—10th scan; solid line—85th scan. The deposited silica-PSS and silica-PDADMAC films had a thickness of ca. 200 µm.

**Figure 2 molecules-30-03246-f002:**
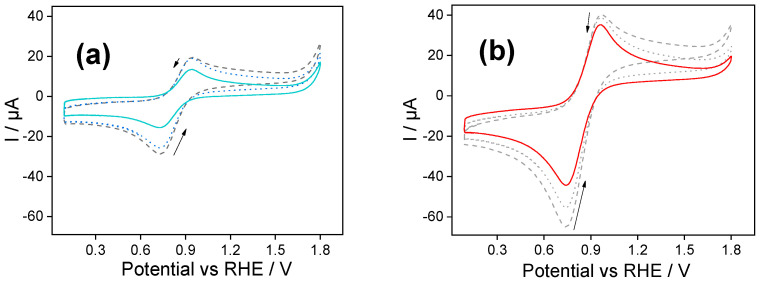
(**a**) Cyclic voltammogram of a Fc@silica-PSS-modified electrode obtained in Trizma buffer. Dashed line—1st scan; dotted line—10th scan; solid line—30th scan. Scan rate: 100 mV s^−1^. (**b**) Repetitive cyclic voltammogram of an Fc@silica-PDADMAC-modified electrode obtained in Trizma buffer solution. Dashed line—first scan; dotted line—25th scan; solid line—50th scan. Scan rate: 100 mV s^−1^.

**Figure 3 molecules-30-03246-f003:**
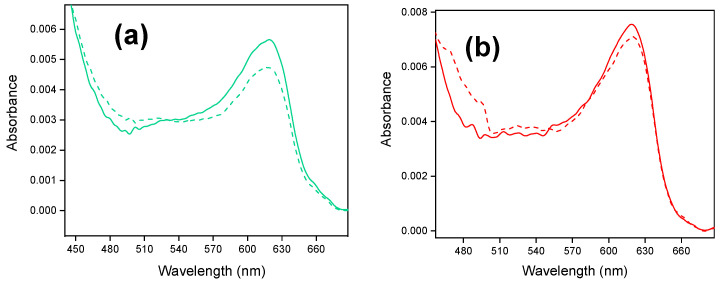
In situ UV-vis spectra of Fc@hybrid silica-modified electrodes: (**a**) for the Fc@silica-PSS-modified electrode, spectra were recorded at +1.42 V (solid line) and +0.22 V (dashed line) vs. RHE; (**b**) for the Fc@silica-PDADMAC-modified electrode, spectra were recorded at +1.42 V (solid line) and +0.22 V (dashed line) vs. RHE.

**Figure 4 molecules-30-03246-f004:**
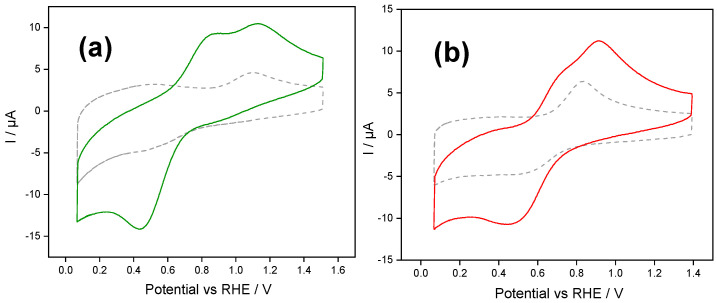
Stabilized cyclic voltammograms of Fc@hybrid silica-modified electrodes in 1 mg mL^−1^ Cyt c + PBS buffer aqueous solution (solid line) and in a Cyt c-free PBS solution (dashed line): (**a**) Fc@silica-PSS-modified electrode and (**b**) Fc@silica-PDADMAC-modified electrode. Scan rate: 100 mV s^−1^.

**Table 1 molecules-30-03246-t001:** Electroactive ferrocene species confined within Fc@silica and Fc@hybrid silica films after stabilization, obtained via cyclic voltammetry (CV) and in situ UV-vis spectroscopy.

Electrode	Electroactive Ferrocene (mM)		Cyt c Charge (µC)
	CV	In Situ UV-Vis	
Fc@silica [17]	0.12	0.19	28
Fc@silica-PSS	0.03	0.23	31
Fc@silica-PDADMAC	0.08	0.11	23

## Data Availability

The original contributions presented in this study are included in the article/Appendix A. Further inquiries can be directed to the corresponding author(s).

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
