# Peer review of "Electroassisted Incorporation of Ferrocene Within Sol–Gel Silica Films to Enhance Electron Transfer—Part II: Boosting Protein Sensing with Polyelectrolyte-Modified Silica"

_molecules, 2025, doi:10.3390/molecules30153246_

Round 1

Reviewer 1 Report

Comments and Suggestions for Authors

The author synthesized Class I hybrid silica matrices incorporating either negatively charged poly(4-styrene sulfonic acid) (PSS) or positively charged poly(diallyl dimethylammonium chloride) (PDADMAC). These hybrid films were deposited onto ITO electrodes and evaluated via cyclic voltammetry in aqueous ferrocenium solutions. The polyelectrolyte charge played a key role in the electroassisted incorporation of ferrocene: silica-PSS films promoted accumulation, while silica-PDADMAC films hindered it due to electrostatic repulsion. These results illuminate the critical roles of electrostatic and hydrophobic interactions in ferrocene incorporation and retention, providing valuable guidance for the design of efficient bioelectrochemical sensors. The manuscript is well-written and professionally presented. Nevertheless, there are several issues in the current version that need to be addressed before it can be considered for publication. The detailed comments are outlined below:

  1. In the Abstract, avoid defining an abbreviation for any term that appears only once; such one-time occurrences do not require acronyms.
  2. Please ensure that citation markers are used consistently throughout the manuscript, and carefully verify that all references conform to the journal’s formatting requirements.
  3. The description of the electrophoretic accumulation process should include voltage, deposition time, and film thickness to ensure reproducibility.
  4. Ensure that units are formatted consistently throughout the manuscript and adhere to the journal’s style guidelines
  5. How do electrostatic and hydrophobic interactions respectively influence the loading and retention of ferrocene in the PSS- and PDADMAC-functionalized silica films?
  6. Briefly explain how cyclic voltammetry was used to determine the amount of electroactive ferrocene within the hybrid films.
  7. Aside from electrostatic attraction and mediator concentration, could the polymer electrolyte modification (PSS or PDADMAC) itself—by altering the local microenvironment of the silica film (e.g., hydrophilicity/hydrophobicity, pore size/tortuosity, interfacial charge distribution)—directly influence the concentration of cytochrome c near the electrode or its effective collision frequency with the embedded ferrocene?

Author Response

We sincerely thank the reviewer for his thoughtful and constructive comments, which have helped us improve the quality and clarity of our manuscript. Below are our point-by-point responses, indicating the revisions made in the updated version of the manuscript.

Response to Reviewer 1

  1. In the Abstract, avoid defining an abbreviation for any term that appears only once; such one-time occurrences do not require acronyms.

Thank you for this valuable observation. We fully agree with your suggestion and have removed all acronyms for terms that appear only once in the Abstract. The full names are now used without abbreviation.

  1. Please ensure that citation markers are used consistently throughout the manuscript, and carefully verify that all references conform to the journal’s formatting requirements.

We appreciate this important comment. The entire manuscript has been thoroughly reviewed to ensure citation markers are used consistently. Additionally, all references have been carefully reformatted to meet the journal’s style and formatting standards.

  1. The description of the electrophoretic accumulation process should include voltage, deposition time, and film thickness to ensure reproducibility.

Thank you for pointing this out. We have now added the missing experimental parameters in the revised manuscript, including new schemes in the supporting information with details on the electroassisted accumulation.

  1. Ensure that units are formatted consistently throughout the manuscript and adhere to the journal’s style guidelines.

We agree with your observation. A thorough review of the manuscript was carried out, and all units have been revised to ensure consistency and adherence to the journal’s formatting standards.

  1. How do electrostatic and hydrophobic interactions respectively influence the loading and retention of ferrocene in the PSS- and PDADMAC-functionalized silica films?

Thank you for this thoughtful and important question. We have updated the manuscript to better clarify the roles of electrostatic and hydrophobic interactions.

Electrostatic interactions primarily influence the initial loading of ferrocene. In silica–PSS films, the negatively charged sulfonate groups (SO₃⁻) attract the positively charged ferrocenium ions, facilitating their effective incorporation. In contrast, the positively charged PDADMAC-modified silica matrix repels ferrocenium ions, thereby limiting their initial loading due to electrostatic repulsion.

Regarding retention, both hydrophobic interactions and redox state play important roles. Although electrostatics hinder the initial loading of ferrocenium (oxidised state) into PDADMAC films, the incorporation of neutral ferrocene (reduced state) is favored by PDADMAC's moderate hydrophobicity. Neutral ferrocene interacts with the hydrophobic backbone of the polymer, allowing it to be retained within the film. Conversely, PSS-based films are more hydrophilic and provide less stabilization for neutral ferrocene, leading to lower incorporation into the film.

These conclusions are supported by cyclic voltammetry and UV–vis spectroscopy, as outlined in the Results and Discussion sections.

Briefly explain how cyclic voltammetry was used to determine the amount of electroactive ferrocene within the hybrid films.

As the reviewer has suggested, we have now included a paragraph in the manuscript describing how cyclic voltammetry was used to estimate the quantity of electroactive ferrocene.

“It should be noted that this value represents an approximation and not the true diffusion coefficient within the film. Therefore, the calculated concentrations are not absolute but serve as relative estimates to compare the retention of ferrocene in the different hybrid silica films. In this context, the peak current intensity (Ip) was obtained from the reduction peak of the stabilized cyclic voltammogram. Using the Randles–Sevcik equation with known parameters (scan rate, electrode area, and estimated diffusion coefficient), we determined the apparent concentration of electroactive ferrocene. From this analysis, the apparent concentration of electroactive ferrocene in the silica-PSS film was approximately 0.03 mM.”

  1. Aside from electrostatic attraction and mediator concentration, could the polymer electrolyte modification (PSS or PDADMAC) itself by altering the local microenvironment of the silica film (e.g., hydrophilicity/hydrophobicity, pore size/tortuosity, interfacial charge distribution), directly influence the concentration of cytochrome c near the electrode or its effective collision frequency with the embedded ferrocene?

We appreciate this insightful comment. As suggested, we have expanded the discussion in the manuscript to address the influence of polyelectrolyte properties on the local microenvironment and their potential effect on cytochrome c behaviour. The revised manuscript now includes the following paragraphs:

“The observed differences in mediated electron transfer between the Fc@silica-PSS and Fc@silica-PDADMAC electrodes can be attributed to variations in surface chemistry and ferrocene retention within the hybrid films. The incorporation of PSS into the silica matrix introduces a high density of negatively charged sulfonate groups, increasing the film’s hydrophilicity and promoting electrostatic interactions with Cyt c in solution. This likely increases the local concentration of Cyt c near the film surface, thereby enhancing the frequency of productive collisions with the confined ferrocene species and facilitating more efficient mediated electron transfer.

In contrast, PDADMAC-functionalized films present a more hydrophobic environment with a lower surface charge density. This modification likely reduces the local concentration of Cyt c near the electrode surface due to weaker electrostatic interactions, limiting its accessibility to the confined ferrocene species within the silica film. Additionally, potential differences in pore morphology or tortuosity introduced by PDADMAC may further hinder Cyt c diffusion or reduce its spatial proximity to electroactive sites.”

Reviewer 2 Report

Comments and Suggestions for Authors

In this paper, the authors presented the results of a study of modified ITO/silica electrodes (ITO/silica-PSS and ITO/silica-PDADMAC). Despite the original idea, the interpretation of the data and the possibility of using these electrodes in analytical chemistry are questionable.

1) The description of the resulting silica-film materials is extremely poor. In particular, the redox response values ​​should significantly depend on the thickness and morphology of the films - these data are not presented in the paper. Also, the appearance of an additional insulating layer should lead to a change in the resistance of the entire electrochemical system, which in turn should affect the shape of the CVA curves. Moreover, the higher the resistance, the more significant the change in the shape of the CVA curves. However, to interpret electrochemical processes, it is better to carry out mathematical processing that takes into account the additional resistance appearing in the system.

It is also worth adding a comparison of the obtained results with the behavior of ferricinium salt using unmodified ITO electrodes.

2) The authors use only a scan rate of 100 mV/s. How does the sweep rate affect the shape of the curves? It is necessary to conduct a series of experiments with different sweep rates.

3) The authors use the Randles-Sevcik equation with the diffusion coefficient of ferricinium cation determined in the solution. This is incorrect. The main electrochemical process occurs in the film, and the diffusion coefficient of ferricinium cation in the film should be used (however, the authors did not determine this value). The authors further discuss the obtained results, but they are obviously incorrect.

4) Reasoning about the processes occurring during CVA should be illustrated by a set of Schemes.

5) Were the authors able to achieve the maximum concentration of ferrocene due to sorption in films (based on the composition of the polymer coating)? Can it be achieved simply by keeping the electrode in a ferricinium salt solution (concentration and increase in response intensity not by performing multiple CVA cycles, but simply by keeping it in the solution - for example, when performing 60 scans, the electrode is kept in the solution for at least 30 minutes (scanning time of 1 cycle is ~30 seconds))?

Thus, the authors described the two insufficiently characterized electrodes. The work requires significant revision and understanding; in the presented form, this publication cannot be published in the Molecules journal.

Author Response

Response to Reviewer 2

  1. The description of the resulting silica-film materials is extremely poor. In particular, the redox response values should significantly depend on the thickness and morphology of the films - these data are not presented in the paper. Also, the appearance of an additional insulating layer should lead to a change in the resistance of the entire electrochemical system, which in turn should affect the shape of the CVA curves. Moreover, the higher the resistance, the more significant the change in the shape of the CVA curves. However, to interpret electrochemical processes, it is better to carry out mathematical processing that takes into account the additional resistance appearing in the system. It is also worth adding a comparison of the obtained results with the behavior of ferricinium salt using unmodified ITO electrodes.

We thank the reviewer for these detailed and constructive comments. In response, we have revised the manuscript to provide a more detailed description of the silica-film materials. Specifically, we have now included information on film thickness based on estimated deposition volume and surface area.

We fully agree that the morphology and structure of films can greatly affect their electrochemical behavior. We consider detailed studies of morphology and porosity as important future research areas to better understand how film structure relates to electrochemical responses.

In the current study, our focus was primarily on the electrochemical performance and mediator incorporation within the silica films. However, to gain additional structural insight, we also conducted nitrogen adsorption (BET) analysis on the dried films. The textural properties of the silica-PSS and silica-PDADMAC samples, derived from N₂ adsorption isotherms, have now been included and discussed in the manuscript.

“To assess the porosity of the hybrid silica materials, nitrogen adsorption analyses were carried out at −196 °C using an automated system (Autosorb-6, Quantachrome). The hybrid silica was synthesized as monoliths using the sol–gel method and subsequently dried under vacuum at 100 °C for 24 hours. Pore size distribution was calculat-ed from the adsorption isotherms using the Density Functional Theory (DFT) model implemented in the Autosorb-6 software (Quantachrome).”

To better understand the structure of the unmodified silica films, we performed Transmission Electron Microscopy (TEM) and Scanning Electron Microscopy (SEM). The TEM image (Figure 1A) showed a dense, amorphous silica with no discernible structured pore network. Similarly, the SEM image (Figure 1B) showed the dried silica gel without defined morphology. These observations confirm the disordered nature of the sol–gel-derived silica matrix and are consistent with the sub-nanometric scale of its internal structure. Furthermore, since the baseline silica matrix does not present clearly defined structural features at this resolution, the presence of polyelectrolytes within the matrix was not evident through TEM or SEM analyses.

Figure R1. Transmission Electron Microscopy (TEM, A) and Scanning Electron Microscopy (SEM, B) images of the sol–gel-derived silica film.

To further investigate the chemical composition of the hybrid silica films, FTIR spectroscopy was performed on bare silica, Silica-PSS, Silica-PDADMAC (see Fig. 2 below). As the reviewer will notice, there are no clear bands linked to PSS or PDADMAC in the spectra of the modified silica films. This is due to the low amount of polyelectrolyte incorporated within the silica network. Indeed the silica bands overlapped with most of the characteristic FTIR signals of the polyelectrolytes. For instance, typical FTIR peaks of PDADMAC include a broad O-H stretch around 3446 cm-1, CH asymmetric and in-plane bending modes near 1635 cm-1 and 1474 cm-1, and a C–N stretch between 1110-1125 cm-1.[2] Similarly, poly(styrene sulfonate) (PSS) exhibits distinctive SO₃⁻ antisymmetric and symmetric stretching bands at 1178  cm-1 and 1037  cm-1, respectively, along with in-plane skeletal and bending vibrations of the benzene ring at 1125  cm-1 and 1008  cm-1.[3]

While these signatures are thoroughly documented for the pure polymers, their absence in our spectra indicates that the polyelectrolytes are incorporated within the silica matrix in quantities that cannot be detected straightforwardly by this technique.

Figure R2. FTIR spectra of pure silica and hybrid silica films.

Since the functionalized monoliths are polyelectrolytes, they are transparent (no turbidity is observed after gelation), which leads us to believe that no phase separation occurs due to the presence of these polyelectrolytes in the silica matrix. This homogeneous distribution allows us to state that the composition of the monoliths is equal to the nominal composition of the precursor sol described in the experimental section.

Concerning the impact of the insulating silica layer on electrochemical behaviour, we recognise the need to account for extra resistance effects caused by film deposition. To address this query, we now present a new Figure S6 (in the Supporting Information) which shows the stabilized cyclic voltammograms recorded for bare ITO, ITO/silica-PSS, and ITO/silica-PDADMAC electrodes in a 0.1 M Trizma buffer solution (pH 8.44).

All three electrodes exhibit a featureless double-layer charging current with no observable faradaic peaks within the studied potential window, confirming the electrochemical stability of the films in the absence of redox-active species.

Notably, the shape of voltammetric profiles (the characteristic box-shape of the capacitative charge processes) indicates that the deposition of silica-polyelectrolyte films does not significantly change the interfacial resistance or impede charge transfer at the ITO surface under these conditions. In cyclic voltammetry, when an electrode has resistance to current flow (i.e., it is poorly conductive), the voltammogram tilts, showing an ohmic-type response. However, no such tilting is observed in these experiments, which suggests that the modification of the ITO electrode with the silica layer does not increase its resistivity. This is because the layer (a hydrogel) is highly porous, and the ionic conductivity through it is similar to that of the solution.

While a more advanced mathematical treatment (e.g., iR compensation) could offer further quantitative insight into resistance contributions, the comparative CV data already suggest that the shifts in voltammetric shape observed in the presence of redox species (Figure 1 in the main manuscript) are mainly driven by mass transport processes rather than resistance effects.

The authors use only a scan rate of 100 mV/s. How does the sweep rate affect the shape of the curves? It is necessary to conduct a series of experiments with different sweep rates.

We would like to clarify that a scan rate study was indeed conducted in our previous work using silica films without polyelectrolyte modification. The resulting cyclic voltammograms at various sweep rates are presented and discussed in that earlier publication.[1]

In the present study, as indicated by the title "Boosting Protein Sensing with Polyelectrolyte-Modified Silica," our objective was to investigate the specific effects of polyelectrolyte functionalization (PSS and PDADMAC) on mediator incorporation and protein-mediated electrochemical responses. To maintain a consistent basis for comparison across the modified films, a single scan rate of 100 mV/s was used throughout the experiments.

However, we agree that scan rate studies can provide additional kinetic insights. Therefore, additional experiments were performed using different scan rates, and the results are presented in the Supplementary Information in Fig. S3 and discussed in the main manuscript in page 5.

“Furthermore, we carried out experiments at varying scan rates, confirming the presence of two different kinetics of heterogeneous electron transfer attributed to mass transport limitations within the hybrid silica films (see Fig. S3 in the Supporting Infor-mation). For both silica-PSS and silica-PDADMAC modified electrodes, the cyclic voltammograms recorded at low scan rates (10-20 mV s–1) displayed a single anodic peak with a bell-shaped profile. This feature is indicative of ferrocene species either adsorbed at the electrode interface or slowly diffusing through the porous silica matrix. Under these conditions, the diffusion layer is sufficiently thick to mask mass transport limita-tions within the films. As the scan rate increased to 50 mV s–1, a noticeable shoulder developed on the anodic wave for both hybrid silica modified electrodes, signaling the emergence of a secondary oxidation process, most likely involving ferrocene species confined deeper in the silica films. At higher scan rates (500 and 1000 mV s⁻1), pro-nounced peak broadening and distortion were evident in both hybrid silica modified electrodes, reflecting growing mass transport limitation and slower heterogeneous elec-tron transfer kinetics across the hybrid silica films.”

  1. The authors use the Randles-Sevcik equation with the diffusion coefficient of ferricinium cation determined in the solution. This is incorrect. The main electrochemical process occurs in the film, and the diffusion coefficient of ferricinium cation in the film should be used (however, the authors did not determine this value). The authors further discuss the obtained results, but they are obviously incorrect.

We agree that the use of a diffusion coefficient determined in solution introduces a simplification, as the electrochemical process occurs within the confined environment of the silica film, where diffusion behavior may differ significantly due to factors such as tortuosity, pore size, and interactions with the film matrix.

In its usual form, the Randles–Sevcik equation applied to planar electrodes depends on two fundamental parameters of the studied redox probe: its diffusion coefficient and its concentration in the bulk solution.

Using this equation with porous electrodes is somewhat more difficult. Although phenomenologically the voltammetric response observed at different scan rates is characteristic (the peak current varies linearly with the square root of the scan rate), in these porous electrodes both the diffusion coefficient and the concentration of the species may be affected by the layer and appear to differ from their values in the bulk solution.

In this work, we have assumed that the diffusion coefficient is not affected by the layer, as we consider it to be a physical constant characteristic of the redox probe that does not change as it passes through the hydrogel layer (which is mostly aqueous solution). Therefore, we treated it as a constant in the Randles–Sevcik equation. However, the layer shows some affinity for the ferrocene probe, and for that reason, the concentration parameter was left as a free variable in the equation, to be quantified.

However, we initially used the diffusion coefficient of ferrocenium in solution as an estimate to compare the apparent concentration of electroactive species within the film. We have revised the manuscript to clarify that the calculated values are approximate and mainly used to compare relative loading and retention between different film types (e.g., PSS vs. PDADMAC).

We are aware of the limitations of this determination, and in later sections we discuss the possibility that the diffusion coefficient may be apparently affected.

  1. Reasoning about the processes occurring during CVA should be illustrated by a set of Schemes.

We appreciate your suggestion. To improve the clarity of the mechanisms and processes discussed in the manuscript, we have added schematic illustrations that summarizes the key steps involved during cyclic voltammetry (CV), including the electroassisted incorporation of ferrocenium species and their subsequent role in mediating electron transfer to cytochrome c (see supplementary information schemes S1, S4, S5 and S7).

  1. Were the authors able to achieve the maximum concentration of ferrocene due to sorption in films (based on the composition of the polymer coating)? Can it be achieved simply by keeping the electrode in a ferricinium salt solution (concentration and increase in response intensity not by performing multiple CVA cycles, but simply by keeping it in the solution - for example, when performing 60 scans, the electrode is kept in the solution for at least 30 minutes (scanning time of 1 cycle is ~30 seconds))?

Thank you for this insightful comment. During our study, we explored various strategies for incorporating ferrocenium species into silica-based films. Beyond the electroassisted method detailed in our manuscript, we also tested passive loading by immersing the electrodes in ferrocenium salt solutions for extended durations, comparable to or even longer than the total time used for the electrochemical accumulation procedure. However, this approach failed to produce any detectable electron transfer response to cytochrome c, indicating insufficient retention or improper positioning of the redox mediator within the film.

Furthermore, we attempted to incorporate the ferrocenium species directly into the sol–gel mixture during the film preparation process. While this strategy initially yielded some level of incorporation, the resulting films demonstrated poor stability. The ferrocenium species were quickly leached out during post-synthesis rinsing and electrochemical cycling, and no mediated electron transfer to cytochrome c could be observed under these conditions.

These findings highlight the importance of the electroassisted method, which seems to enable more thorough and stable confinement of the redox mediator within the film. This approach likely benefits from both electric-field-driven transport and an improved distribution of ferrocene species in accessible regions of the porous matrix.

It is clear from the work that it is necessary for the ferrocenium to undergo a redox change to ferrocene. The change from a +1 charged species to a neutral one promotes the stable incorporation and retention of ferrocene within the inorganic matrix.

Consequently, our results suggest that changing the redox state of ferrocene during accumulation is not supplementary but crucial for stable mediator retention and efficient electron transfer to cytochrome c.

Reviewer 3 Report

Comments and Suggestions for Authors

This paper investigates the electroassisted incorporation of ferrocene into Class I hybrid silica films modified with polyelectrolytes (PSS or PDADMAC) to enhance mediated electron transfer for protein sensing, specifically cytochrome c (Cyt c). The authors use sol–gel methods to incorporate negatively charged PSS or positively charged PDADMAC into silica films on ITO electrodes. They characterize the films using cyclic voltammetry (CV) and in situ UV–vis spectroscopy to assess ferrocene accumulation and retention. They found PSS films promote ferrocene accumulation due to electrostatic attraction, while PDADMAC films hinder it due to repulsion. However, the current form of the manuscript lacks much critical information. Therefore, a significant revision is required to sharpen the focus, increase the depth of discussion, improve formatting, and enhance visual presentation.

  1. The concept of using polyelectrolytes to modulate charge interactions in silica films is not new (e.g., Walcarius’ prior work, https://doi.org/10.1002/tcr.202300194). The paper does not clearly articulate how this work advances beyond existing literature.
  2. Film thickness, porosity, and surface roughness are not measured (e.g., ellipsometry, SEM, AFM). These properties critically affect diffusion and electron transfer.
  3. Polyelectrolyte loading is not quantified (e.g., via XPS or elemental analysis), making it impossible to correlate charge density with ferrocene accumulation.
  4. There is no comparison with the physical adsorption of ferrocene (without electroassistance) to isolate the role of the applied potential.
  5. No pH-dependent studies to probe the role of polyelectrolyte charge (PSS pKa = 1.5, PDADMAC pKa = 10.5) near neutral pH.
  6. Cyt c adsorption artifacts: Cyt c (pI ~10.5) is positively charged at pH 7.3, but the paper does not account for non-specific adsorption to PSS films, which could inflate the mediated signal.
  7. The system is tested in idealized buffer solutions with high Cyt c concentrations (1 mg/mL). Performance in serum or complex matrices is not evaluated, leaving it without real-world validation.
  8. While leaching is mentioned (Figure 2), long-term stability (days/weeks) under continuous cycling is not assessed.
  9. The authors should enhance the article's visual impact and accessibility. For example, the authors should design a concise schematic figure to summarize key achievements and the conceptual design for this paper. This would help readers quickly retain the key messages.
  10. The flow of this article is terrible. Section 3.1 was only the sub-paragraph in section 3. What is the meaning of dividing section 3 if only one sub-section exists?

Recommendations for Improvement

  1. Quantify film properties (thickness, charge density, porosity) and correlate with ferrocene loading.
  2. Use a rotating disk electrode (RDE) or microelectrode arrays to decouple diffusion from electron transfer kinetics.
  3. Include pH-dependent studies to validate polyelectrolyte charge effects.
  4. Compare with covalently bound ferrocene (Class II hybrids) to highlight the advantages of electrostatic accumulation.
  5. Test real samples (e.g., serum, cell lysates) and assess interference from other proteins.
  6. Use electrochemical impedance spectroscopy (EIS) to quantify electron transfer kinetics for Cyt c.

Author Response

Response to Reviewer 3

  • The concept of using polyelectrolytes to modulate charge interactions in silica films is not new (e.g., Walcarius’ prior work, https://doi.org/10.1002/tcr.202300194). The paper does not clearly articulate how this work advances beyond existing literature

Thank you for this valuable comment. We fully acknowledge that the use of polyelectrolytes to modulate charge interactions in silica films has been previously explored, including in the work of Walcarius and co-workers [e.g., https://doi.org/10.1002/tcr.202300194]. However, our study makes a distinct and meaningful contribution in several key aspects:

Previous approaches often rely on covalent modification or post-synthetic grafting to immobilize redox-active species within silica matrices, typically using organosilane chemistry or click-chemistry reactions. While effective, these methods involve multistep syntheses, careful reagent design, and often require specific reaction conditions. In contrast, our work introduces a simple, one-step electroassisted loading strategy that exploits redox-driven accumulation to incorporate ferrocene into sol-gel-derived, polyelectrolyte-containing silica films.

 We thoroughly compared our electroassisted accumulation method to passive loading and in situ incorporation during film formation. Both alternatives failed to produce stable retention of ferrocene or effective electron mediation to cytochrome c, likely due to poor confinement or unfavorable localization of the redox species. These comparisons highlight that the redox-state switching involved in the electroassisted process is essential for achieving stable immobilization and functional electron transfer capabilities.

Unlike prior studies, we do not treat the polyelectrolyte as a mere additive. Instead, we conduct a comparative investigation of two oppositely charged polyelectrolytes (PSS and PDADMAC) and demonstrate their markedly different impacts on ferrocene retention, redox stability, and electron transfer to cytochrome c. This direct side-by-side evaluation provides new mechanistic insights into how electrostatic interactions and matrix composition govern redox mediator behavior in such systems.

While much of the earlier literature focuses on Class II materials that feature covalently integrated organic components, we demonstrate that Class I hybrids, assembled through purely electrostatic and physical interactions, can offer sufficient mediator stability and redox activity when properly engineered using our method. This finding expands the scope and utility of Class I materials, offering a more accessible platform for bioelectrochemical applications

To address this concern, we have revised the Introduction section to better emphasize how this work contributes new insights into the interplay between polyelectrolyte charge, hydrophobicity, and redox protein detection within hybrid silica films.

  • Film thickness, porosity, and surface roughness are not measured (e.g., ellipsometry, SEM, AFM). These properties critically affect diffusion and electron transfer.

We thank the reviewer for this important comment. We fully agree that film thickness, porosity, and surface roughness are critical parameters influencing diffusion and electron transfer in nanostructured films. While the primary aim of our study was to evaluate electrochemical behavior and redox mediation efficiency in polyelectrolyte-containing silica films, we have made efforts to address structural characteristics as follows:

We estimated the film thickness from the known deposition volume and geometric surface area of the electrode. These estimates have been added to the revised manuscript to provide a basic sense of film dimensions. Nitrogen adsorption/desorption measurements (BET analysis) were carried out on the dried silica-PSS and silica-PDADMAC samples to obtain information about their specific surface area and porosity. The resulting data are now included and discussed in the main manuscript.

To further investigate the morphology of the unmodified silica films, we conducted both Transmission Electron Microscopy (TEM) and Scanning Electron Microscopy (SEM) analyses. The TEM micrograph (Figure 1A) revealed a dense, amorphous silica structure with no clearly resolvable pore network. Similarly, the SEM image (Figure R1B) showed smooth, featureless surfaces without distinct morphological pores. These observations confirm the disordered nature of the sol–gel-derived silica matrix and are consistent with the sub-nanometric scale of its internal structure. Furthermore, since the baseline silica matrix does not present clearly defined structural features at this resolution, the presence of polyelectrolytes within the matrix is also not readily detectable by TEM or SEM. The low contrast and distribution of organic functional groups within the dense matrix limit their visualization using these techniques.

Figure R1. Transmission Electron Microscopy (TEM, A) and Scanning Electron Microscopy (SEM, B) images of the sol–gel-derived silica film.

To further investigate the chemical composition of the hybrid silica films, FTIR spectroscopy was performed on bare silica, Silica-PSS, Silica-PDADMAC (The latter spectra are presented in Figure R2). However, no distinct peaks attributable to PSS or PDADMAC were detected in the spectra of the modified silica films. This is most likely due to the relatively low polyelectrolyte content and their physical confinement within the dense silica network, which may reduce their infrared signal intensity and obscure characteristic features. For reference, the typical FTIR peaks of PDADMAC include a broad O-H stretching vibration around 3446 cm-1, CH asymmetric and in-plane bending modes at approximately 1635  cm-1 and 1474  cm-1, respectively, and a C–N stretching vibration in the range of 1110-1125  cm-1.[2] Similarly, poly(styrene sulfonate) (PSS) exhibits distinctive SO₃⁻ antisymmetric and symmetric stretching bands at 1178  cm-1 and 1037  cm-1, respectively, along with in-plane skeletal and bending vibrations of the benzene ring at 1125  cm-1 and 1008  cm-1.[3] While these signatures are well-documented for the pure polymers, their absence in our spectra suggests that the polyelectrolytes are embedded within the silica matrix in amounts or environments that limit their spectroscopic visibility.

Figure R2. FTIR spectra of pure silica and hybrid silica films.

  • Polyelectrolyte loading is not quantified (e.g., via XPS or elemental analysis), making it impossible to correlate charge density with ferrocene accumulation.

We thank the reviewer for this valuable observation. The silica films in this study were prepared using the sol–gel method, with polyelectrolytes incorporated at a controlled and sufficiently high concentration (0.5 mM). The impact of these polyelectrolytes on film properties (particularly on charge distribution and mediator behaviour) was clearly reflected in the recorded cyclic voltammograms, which showed distinct differences in ferrocene loading and retention between PSS- and PDADMAC-modified silica films.

Since the functionalized monoliths are polyelectrolytes, they are transparent (no turbidity is observed after gelation), which leads us to believe that no phase separation occurs due to the presence of these polyelectrolytes in the silica matrix. This homogeneous distribution allows us to state that the composition of the monoliths is equal to the nominal composition of the precursor sol described in the experimental section.

While we acknowledge that techniques such as XPS or elemental analysis could provide a more quantitative assessment of polyelectrolyte loading and surface charge density, we did not consider these analyses essential within the scope of this study, as our focus was on comparative electrochemical performance. Nevertheless, we agree that such complementary characterization would be valuable for further understanding structure/function relationships, and we plan to include it in future work.

  • There is no comparison with the physical adsorption of ferrocene (without electroassistance) to isolate the role of the applied potential.

We thank the reviewer for raising this important point. As noted in our response to a previous reviewer, beyond the electroassisted method presented in the manuscript, we attempted passive loading by immersing the electrodes in ferrocenium salt solutions for prolonged periods, comparable to or exceeding the total time of electrochemical accumulation. This method, however, did not lead to any observable electron transfer to cytochrome c, suggesting poor retention or unfavorable spatial distribution of the mediator within the film.

We also explored the direct incorporation of ferrocenium into the sol-gel precursor solution during film preparation. While this approach allowed some initial loading, the incorporated species were not stably confined and underwent significant leaching during rinsing or electrochemical cycling. Consequently, no mediated electron transfer to cytochrome c was detected.

It is clear from the work that it is necessary for the ferrocenium to undergo a redox change to ferrocene. The change from a +1 charged species to a neutral one promotes the stable incorporation and retention of ferrocene within the inorganic matrix.

These findings clearly highlight the advantages of the electroassisted approach, which promotes more effective confinement of the redox mediator, likely through electric-field-driven transport and favorable localization within the electroactive regions of the porous matrix. As such, our results demonstrate that switching the redox state of ferrocene during accumulation is not merely incidental, but essential for ensuring stable retention of the mediator and enabling efficient electron transfer to cytochrome c.

No pH-dependent studies to probe the role of polyelectrolyte charge (PSS pKa = 1.5, PDADMAC pKa = 10.5) near neutral pH.

We thank the reviewer for this insightful comment. We fully agree that a systematic exploration of pH-dependent behaviour could offer important insights into the role of polyelectrolyte charge states on protein interaction and redox mediator performance. In this study, we focused primarily on evaluating the electrochemical behaviour of polyelectrolyte-functionalized silica films at near-physiological pH (~7.3), which is relevant for the final application of the material (direct electrochemistry to redox proteins) and potential bioanalytical and biosensing applications involving proteins such as cytochrome c.

We note, however, that exploring pH effects in this system presents certain limitations. The isoelectric point (IEP) of the silica matrix is around pH 2, meaning that below this value, silica loses its negative surface charge. Similarly, the sulfonate groups in PSS (pKa ≈ 1.5) remain deprotonated and negatively charged above this pH, but protonation, and thus charge neutralization would only occur under extremely acidic conditions (pH < 1.5). Also, measurement at high pH may produce instabilities in the silica matrices due to its solubility in alkaline media. Operating under such extreme pH conditions could compromise the integrity of the silica network and affect the stability of both the film and the protein analyte.

Given these considerations, we opted to conduct all electrochemical and interaction studies at pH 7.3, where both the silica matrix and PSS maintain their expected charge characteristics, and cytochrome c remains positively charged (pI ≈ 10.5), enabling meaningful evaluation of electrostatic interactions. Nonetheless, we recognize the value of conducting more detailed pH-dependent studies and will consider this in future work.

  • Cyt c adsorption artifacts: Cyt c (pI ~10.5) is positively charged at pH 7.3, but the paper does not account for non-specific adsorption to PSS films, which could inflate the mediated signal.

We appreciate this valuable observation regarding potential cytochrome c (Cyt c) adsorption artifacts. To address this point, we conducted a control experiment to assess possible non-specific interactions between Cyt c and the silica-PSS film. Figure S7 presents the stabilized cyclic voltammograms of the silica-PSS-modified electrode recorded in PBS buffer (pH 7.4), both in the absence and presence of 1 mg mL-1 Cyt c. In both cases, no redox peaks were observed, confirming the absence of direct electron transfer and suggesting that any adsorption of Cyt c onto the film does not result in an electroactive signal.

While electrostatic attraction between the positively charged Cyt c (pI ≈ 10.5) and the negatively charged sulfonate groups of PSS could occur at this pH, such interaction does not appear to translate into measurable electrochemical activity. This is likely due to the small size of Cyt c (~3 nm) relative to the structural characteristics of the mesoporous silica network, which may limit its effective diffusion or orientation at the electrode interface. Taken together, these findings suggest that non-specific adsorption of Cyt c is not a significant contributor to the observed mediated electron transfer response.

  • The system is tested in idealized buffer solutions with high Cyt c concentrations (1 mg/mL). Performance in serum or complex matrices is not evaluated, leaving it without real-world validation.

We appreciate your valid observation. In this study, our primary aim was to investigate the fundamental interactions between the hybrid silica films, the embedded redox mediator (ferrocene), and cytochrome c under controlled conditions. For this reason, we chose to evaluate the system in buffer solution using relatively high concentrations of cytochrome c to ensure a clear and measurable response that could be directly attributed to film composition and mediator behaviour.

We agree that testing in complex biological matrices such as serum is essential for real-world validation. We consider this work as a proof-of-concept platform for further development, and future studies will focus on assessing selectivity, stability, and analytical performance in biologically relevant environments.

  • While leaching is mentioned (Figure 2), long-term stability (days/weeks) under continuous cycling is not assessed.

Thank you for raising this important point. While our primary focus was on demonstrating the efficiency and selectivity of the electroassisted incorporation of ferrocene species, we also performed preliminary stability tests over extended periods. Specifically, once the silica-modified electrodes were prepared, they were stored for several days at 4 °C in Trizma buffer solution. Cyclic voltammetry measurements were then carried out periodically over the course of several days. The recorded voltammograms remained consistent, showing no significant decline in electrochemical signal, which suggests that the incorporated redox mediator remains confined within the silica matrix over time. These findings provide an initial indication of the system’s robustness, although more comprehensive long-term cycling studies could be explored in future work.

The leaching experiments presented in Figure 2 were specifically designed to probe short-term retention behaviour and reproducibility across repeated voltammetric cycles immediately after mediator incorporation. These experiments provided mechanistic insight into the interplay between electrostatic and hydrophobic interactions in film design, which directly affect the stability of the confined redox species.

We agree that extended testing over days or weeks under continuous operation would provide additional information on film durability, and we consider this an important next step.

  • The authors should enhance the article's visual impact and accessibility. For example, the authors should design a concise schematic figure to summarize key achievements and the conceptual design for this paper. This would help readers quickly retain the key messages.

Thank you for this valuable suggestion. In response, we have included a schematic illustration that outlines the key stages of the cyclic voltammetry (CV) process (see schemes in the supplementary information), highlighting the electroassisted incorporation of ferrocenium species and their role in facilitating electron transfer to cytochrome c. We believe this visual addition significantly enhances the clarity and accessibility of the work.

  • The flow of this article is terrible. Section 3.1 was only the sub-paragraph in section 3. What is the meaning of dividing section 3 if only one sub-section exists?

We thank the reviewer for this helpful observation. In response, we have revised the structure of Section 3 by adding a subsection titled (Electrochemical Performance of Fc@ hybrid Silica-Modified Electrodes for Electron Transfer to Cyt c). This addition improves the logical flow of the Results and Discussion section and justifies the division into multiple sub-sections, enhancing the overall readability and organization of the manuscript

We sincerely thank you for the constructive and insightful recommendations. Each of the suggested directions represents a valuable opportunity to strengthen and expand the present work.

Below we provide our detailed responses to each point:

  1. Quantify film properties (thickness, charge density, porosity) and correlate with ferrocene loading

We agree that correlating these structural parameters with ferrocene loading would provide deeper mechanistic insight. While direct measurements were not performed in this study, we ensured that film fabrication was carried out under controlled and reproducible conditions. This point will be considered as a priority in future work, potentially using profilometry, zeta potential, or porosimetric techniques.

  1. Use a rotating disk electrode (RDE) or microelectrode arrays to decouple diffusion from electron transfer kinetics

We fully acknowledge the value of this approach for decoupling mass transport effects from intrinsic electron transfer behaviour. While RDE or microelectrode experiments were not conducted in the current work, we have taken care to analyse the electrochemical data under diffusion-limited conditions and maintain consistent parameters (e.g., scan rate, concentrations) across all electrode systems to allow for comparative analysis.

  1. Include pH-dependent studies to validate polyelectrolyte charge effects

As noted in our response to Comment 4, this is indeed an important point. In the present study, experiments were performed at physiological pH (7.4) to reflect biologically relevant conditions. However, we agree that a systematic pH-dependent study would help clarify the impact of polyelectrolyte charge states on protein and mediator behavior. This will be a key focus in follow-up work.

  1. Compare with covalently bound ferrocene (Class II hybrids) to highlight the advantages of electrostatic accumulation

This is an excellent suggestion for establishing a clearer performance benchmark. Although Class II hybrid systems were not included in this study, the advantages of electrostatic accumulation were assessed through the relative comparison of mediator response and protein activity across differently functionalized films. A direct comparison with covalently bound systems will be pursued in future studies.

  1. Test real samples (e.g., serum, cell lysates) and assess interference from other proteins

We appreciate this suggestion to evaluate real-world applicability. Our current goal was to develop and characterize a proof-of-concept sensing platform under ideal conditions using Cyt c as a model protein. Testing in complex biological matrices will be a crucial step in future validation and application of this system.

  1. Use electrochemical impedance spectroscopy (EIS) to quantify electron transfer kinetics for Cyt c

We agree that EIS could provide complementary kinetic information. In this study, we focused on cyclic voltammetry to evaluate redox behavior and protein–film interactions. Future work will aim to incorporate EIS measurements to quantitatively assess charge transfer dynamics.

We would like to thank you again for your thoughtful input, which will guide our ongoing and future research efforts.

[1]      R.I. Loughlani, A. Gamero-Quijano, F. Montilla, Electroassisted Incorporation of Ferrocene within Sol–Gel Silica Films to Enhance Electron Transfer, Molecules. 28 (2023) 6845. https://doi.org/10.3390/molecules28196845.

[2]      S. Tan, S. Jiang, X. Li, Q. Yuan, Factors affecting N-nitrosodimethylamine formation from poly(diallyldimethylammonium chloride) degradation during chloramination, R. Soc. Open Sci. 5 (2018). https://doi.org/10.1098/rsos.180025.

[3]      P. Balding, R. Borrelli, R. Volkovinsky, P.S. Russo, Physical Properties of Sodium Poly(styrene sulfonate): Comparison to Incompletely Sulfonated Polystyrene, Macromolecules. 55 (2022) 1747–1762. https://doi.org/10.1021/acs.macromol.1c01065.

Round 2

Reviewer 1 Report

Comments and Suggestions for Authors

The manuscript has been well revised and meets the publication requirements.

Reviewer 2 Report

Comments and Suggestions for Authors

The authors have better explained the essence of the article after the review . There are a number of debatable issues. In particular, the application of the Randles-Sevcik equation. This equation is used for two soluble redox forms of the compound, while ferrocene is not a soluble compound in water and must precipitate in micropores. I hope that the authors will pay attention to this in the future.

This publication can be published in the presented form

Reviewer 3 Report

Comments and Suggestions for Authors

The authors have made substantial revisions. Now it can be published.